# Reflections on Digital Maternal and Child Health Support for Mothers and Community Health Workers in Rural Areas of Limpopo Province, South Africa

**DOI:** 10.3390/ijerph20031842

**Published:** 2023-01-19

**Authors:** Livhuwani Muthelo, Masenyani Oupa Mbombi, Mamare Adelaide Bopape, Tebogo M. Mothiba, Melissa Densmore, Alastair van Heerden, Shane A. Norris, Nervo Verdezoto Dias, Paula Griffiths, Nicola Mackintosh

**Affiliations:** 1Department of Nursing, University of Limpopo, Polokwane 0727, South Africa; 2Department of Computer Science, University of Cape Town, Cape Town 7700, South Africa; 3Centre for Community Based Research, Human Sciences Research Council, Pietermaritzburg 3201, South Africa; 4SAMRC/Wits Developmental Pathways for Health Research Unit (DPHRU), Faculty of Health Sciences, University of the Witwatersrand, Johannesburg 2198, South Africa; 5School of Computer Science and Informatics, Cardiff University, Cardiff CF24 4Ag, UK; 6School of Sport, Exercise and Health Sciences, Loughborough University, Ashby Road, Loughborough LE11 3TU, UK; 7Department of Population Health Sciences, College of Life Sciences, University of Leicester, George Davies Centre, University Road, Leicester LE1 7RH, UK

**Keywords:** digital health, mother and child health, mothers

## Abstract

Introduction: Digital health support using mobile and digital technologies, such as MomConnect and WhatsApp, is providing opportunities to improve maternal and child healthcare in low- and middle-income countries. Yet, the perspective of health service providers, pregnant women, and mothers as recipients of digital health support is under-researched in rural areas. Material and Methods: An exploratory-descriptive qualitative research approach was adopted to reflect on the experiences of mothers, community leaders, and community health workers on mobile health opportunities in the context of maternal and child health in rural areas. Purposive sampling was used to select 18 participants who participated in the two focus groups and individual semi-structured interviews for data collection about digital maternal and child health support. The thematic open coding method of data analysis assisted authors in making sense of the given reflections of mothers, community leaders, and healthcare workers about digital health support. Results: Participants commented on different existing digital support apps and their importance for maternal and child health. For example, MoMConnect, Pregnancy+, WhatsApp, and non-digital resources were perceived as useful ways of communication that assist in improving maternal and child health. However, participants reported several challenges related to the use of digital platforms, which affect following the health instructions given to pregnant women and mothers. Conclusions: Participants expressed the significant role of digital support apps in maternal and child health, which is impacted by various challenges. Addressing the lack of digital resources could improve access to health instructions for pregnant women and mothers.

## 1. Introduction and Background

In 2015, all countries, guided by the United Nations (UN), committed to reducing the maternal and child mortality rate to less than 70 per 100 live births by 2030 [1]. Although there is some progress in the rate of maternal and child mortality, every 11 s a pregnant woman or newborn dies somewhere in the world, accounting for 2.8 million deaths in a year [1]. This indicates a global concern regarding the increased rate of maternal deaths that could be preventable and treatable using digital support [2,3]. For example, Southern Asia, which has not witnessed digital improvements for mobile health support, faces a disturbing situation of stillbirths and maternal and neonatal deaths as compared to East Asia [3]. For example, China is one of the countries in East Asia that demonstrates improved mobile health support for maternal and child health by recording a total of 5276 Android apps and 877 iOS apps developed for the digital support of maternal and child healthcare [4]. We believe that by adopting the digital improvement strategies from China, most African countries can potentially improve the high number of maternal and child deaths.

The use of mobile health technologies has benefited healthcare users and patients across the globe [5]. These benefits include increasing the accessibility of medical healthcare practitioners to patients and an easier way of communicating maternal and child health messages [6]. Another benefit is that community members in disadvantaged areas can have rapid access to convenient maternal and child health services and improved antenatal and maternal health as supported by digital resources [6,7,8]. Several authors have reported mHealth as a significant resource to help track pregnancies and improve maternal health, such as antenatal care in low- and middle-income countries (LMICs) [9,10]. On the other hand, malnutrition has been identified as a major concern in developing countries; therefore, the use of digital health support could be a way to improve access to preventive measures for malnutrition among children [10].

Digital health support is support provided via digital channels such as email, chat, mobile apps (mHealth), social media, and more to enable mothers and community health workers to interact with healthcare providers using their smartphones, tablets, and computers [11]. While previous research has highlighted the potential of digital health support to improve maternal health, Mehl [8] stated that digital support utilizing mobile and wireless technologies for health could also intensify inequalities, especially in supporting disadvantaged rural communities [12,13]. The high number of maternal deaths in some sub-Saharan Africa reflects high inequalities in access to quality health services and the gap between rich and poor, which can potentially be alleviated by providing digital support [1,14]. The same issue was also raised in 2017 by the International Telecommunication Union (ITU), that Africa was at the tail of ICT usage in the world, with just above 20% of the population having access to the Internet, and over 20% without access even to basic voice communications [15]. However, most people living in rural communities face cultural, economic and social barriers to communicating through it. While in South Africa, historically, rural communities have been underserved in terms of telecommunication services due to inequality between those who have the skills and financial resources to use the Internet optimally and those who do not [16]. To curb this, the State of Broadband Catalyzing Sustainable Development 2016 has recommended the use of a community networks model (a free, open and mostly wireless telecommunications community network) to enable historically disadvantaged communities, many of which are in rural areas, to obtain the technical and related skills, including support, to install, manage and operate their own electronic communications services and infrastructure [16]. Although different African countries continue to thrive for digital support to improve maternal and child health and as part of a strategic approach to expanding health services, there is a need for defining a standard framework for implementation across the board [17]. For example, the Ondo State in Nigeria implemented the Abiye Safe Motherhood Initiative to reduce maternal deaths, while a start-up in Cameroon designed the GiftedMom app to prevent complications during childbirth [18]. South Africa launched a maternal mHealth project to improve maternal, neonatal, and child health problems. However, though these initiatives have demonstrated the potential to curb the high maternal deaths, there is still a need for digital support targeted to improve child health, which includes reflections on the digital initiatives for improving end-user experiences. The above example indicates that the African continent prioritizes improving maternal and child health complications by providing digital support such as the Safe Delivery app in Ethiopia and Zero Mothers Die app in Ghana, Gabon, Mali, Nigeria, South Africa, and Zambia [18,19]. The WHO Compendium of Innovative Health Technologies for Low Resource Settings revealed that there is still a gap in terms of digital health training in order to ensure technological proficiency [20]. The regularity and frequency of training will probably have an impact on users’ confidence and competency, including patients and providers [17]. Previous studies have also identified challenges related to capacity issues, where patients and community health workers were unable to use their mobile devices and online documents as needed, which resulted in the retrieval of information that was out of date and erroneous [21,22]. More importantly, it was also outlined that in low-resource communities, the relationship between healthcare providers and patients can be improved by having literacy in digital communication. This connection may improve understanding and treatment adherence, making it novel in the development of the mHealth system [17].

South Africa has made significant progress in reducing maternal and child mortality rates in the last two decades, but there is still a need to improve maternal and child healthcare services to reach Sustainable Development Goals (SDG) 4 and 5 by 2030 [18]. The literature in the South African context has revealed different perspectives regarding the use, design, and development of digital support for maternal and child health [18,19] which saw it being ranked the highest in Africa. However, despite the good ranking, there is still a gap in developing and improving mobile health digital support for maternal and child health in comparison to other countries such as China [13] and the Global North. In response to some of the digital gaps in improving maternal health services, the National Department of Health (NDoH), 2014 in South Africa, has initiated digital health apps such as MomConnect through the use of cell phones [7]. In 2019, the National Digital Health Strategy for South Africa 2019–2024 was also aimed at improving the health of all South Africans, which is supported by digital health [23]. Moreover, digital health is expected to be a significant driver of health system transformation, and also advance the national Department of Health’s vision of ‘A long and healthy life for all South Africans’ [23]. These initiatives’ targets were to improve communication and information access for pregnant and breastfeeding women through registration to maternal health databases, chat rooms, and educational programs. The literature has revealed that in South Africa, mobile health apps such as MomConnect have so far played a great role in reducing the burden of maternal morbidity [8,24]. However, the use and benefits of these digital applications are non-dominant in rural communities due to socioeconomic and technological reasons [20]. Furthermore, another advantage of mobile applications is that they can support learning, reflective practices and provide emotional support for midwives working in remote and resource-poor areas [25]. Community health workers in South Africa, who offer a range of community-based services, have started using mobile technologies to support a more accurate collection of data during visits, screening practices, and communication with women [26,27,28]. In SA, community health workers’ major role is to serve and respond to the community’s health needs. They do not receive formal professional training or a certificated degree level of education, their training is specific to the intervention [29]. The current study was conducted in rural and poor communities, where health services are invariably inadequate. Therefore, the main responsibility of CHWs is to work at the local level to promote health services and help the community become more knowledgeable about health priorities. Their unique importance comes from the fact that they are rooted in the communities they serve as a person’s first point of contact with the primary health care (PHC) system. Moreover, the literature revealed that in SA, CHWs preventive and promotive programs have contributed to dramatic reductions in infant mortality rates [29].

Various authors continue to deliberate on the opportunities of mobile apps as digital support for improving maternal and child health services in developing regions, especially in rural areas [30,31,32]. Islam et al. also argued that the use of smartphone applications and psychoeducational interventions provided by community health workers (CHWs) might assist in reducing the burden of maternal health challenges, especially in low- and middle-income countries (LMICs) [33]. Whereas Kabongo et al. posit that in LMICs the use of mHealth is determined by various factors such as socio-cultural characteristics, socio-economic, network infrastructure, and connectivity and awareness [34]. As such, a study carried out by Bonciani, Rosis, and Vainieri on mobile health and intervention in maternal care also recommended that further research be conducted on the evaluation of mHealth interventions to identify gaps and provide useful insights for supporting the introduction of mobile-based innovations in maternal and newborn care pathways in other areas, both in developed and developing countries [35]. While noting that South Africa is a middle-income country with socio-economic inequalities that affect resource and service distribution, there remain concerns about high levels of maternal and child morbidity and mortality in deep rural areas that could be related to inequality in accessing healthcare services related to the geography and low socio-economic status of these communities [24,36]. This suggests the need to explore and prioritize the provision of digital support in these rural areas dominated by inequalities. To address challenges in rural areas, digital app developers and network designers ought to obtain baseline information about digital services within rural areas. Therefore, against this background, this study aims to explore the perspectives of mothers, community health workers, and community leaders on maternal and child health digital support in rural, disadvantaged areas of South Africa. In addressing this, the following questions were central to our discussion in addressing this aim:What are the experiences on the use of different existing digital support apps for maternal and child health?What are the challenges encountered by mothers during the antenatal and postnatal period?What can be done to improve the use of digital maternal and child health?

## 2. Material and Methods

The project adopted a qualitative exploratory-descriptive research approach to study digital health support for mothers and community health workers about maternal and child health in rural areas of Limpopo Province in South Africa. According to Maxwell (2013), qualitative research focuses on understanding meanings, motives, aspirations, beliefs, values, and attitudes, which relate to a deeper realm of relationships, processes, and phenomena [37]. As such, the exploratory-descriptive research designs assisted the authors in obtaining in-depth reflections from mothers and community health workers regarding digital health.

### 2.1. Ethical Approval and Informed Consent Statement

Permission was obtained from the University of Limpopo ethics committee (TREC:264/2020:IR), the Department of Health Limpopo Province, and the Cardiff School of Computer Science and Informatics ethics committee. Information was provided on the nature of the research and its demands on participants. The participants were also informed about their right to decide to terminate their participation in the study upon giving written consent. Consent was obtained.

### 2.2. Study Setting

The study was conducted in the tribal community halls of three rural villages outside Polokwane City in the Limpopo Province of South Africa. The three rural villages include Dikgale, Nobody-Ga-Mothiba, and J-Mamabolo, predominantly rural areas with primary healthcare services that include maternal and child health.

### 2.3. Population and Sampling

The population comprises eighteen participants from the three rural areas (Dikgale, Nobody-Ga-Mothiba, and J-Mamabolo) in the Capricorn District, Limpopo Province. The three rural areas selected have a high unemployment rate, poor road infrastructure, and poor health service delivery. The three (3) villages are zoned as rural, low-income areas with a monthly income of R12,508 and are located 90 km east of Polokwane. This community is representative of the Sepedi people, with 95% of the 9353 people in this population listed as speaking Sepedi. Additionally, Kyei [38] and Mothiba et al. [39] indicated concern about a high rate of teenage pregnancy in deep rural areas of Limpopo, which suggests a need for improving maternal and child healthcare services for this most vulnerable group in the area. A purposive sampling technique was applied to select eighteen participants, including mothers, community health workers, and community leaders (who were fathers as well), who were interviewed from each village [40]. Table 1 below illustrates the 18 participants from the three villages who participated in the study.

Participation in the study was voluntary, and they were allowed to terminate participation in the study at any time without any retribution. The three categories were selected based on their significant roles in community service delivery, including health services. For example, the community leaders (male figures also assuming the role of fathers) are part of the political structure in the community that ensures good health service delivery to all; therefore, obtaining their perspective on digital support as service delivery to pregnant women and mothers plays a necessary role for the study’s aim. While the community healthcare workers are the mediators for service delivery between the health service centers and the community members, their reflections are important to understand and improve digital support for pregnant women and mothers. Lastly, mothers are the sole recipients of maternal and child health services, so obtaining their reflections on digital support becomes a significant point of reference. The inclusion criteria were only mothers above 18 years old with children less than two years old. Community healthcare workers needed to be in contact and assist in caring for mothers and children less than two years daily to be included. We only selected community leaders who are fathers with at least a background in the community and their needs, including maternal and child health services. All the participants were purposively selected and data collection was guided by data saturation which was achieved by obtaining rich and concise information on the lived experiences of eighteen participants, and additional sampling offered replication of the information about maternal and child digital support [41].

### 2.4. Data Collection

The recruitment of the participants for data collection was conducted telephonically and face-to-face with the four community leaders to secure a meeting on convenient dates and times. The rationale for using face-to-face meetings and interviews during COVID-19 was due to poor internet connectivity in the rural area. The arrangement of the meeting assisted in indicating the objectives of the study before commencing with data collection. The first four authors conducted a workshop before data collection to build rapport with the participants and describe the study’s rationale before obtaining consent from the participants. The authors also briefed the participants on the interview procedure, risks, and benefits of participating in the study and preparing them for interviews. More importantly, COVID-19 safety rules and regulations such as sanitizing, wearing masks, and social distancing were enforced during data collection to protect researchers and participants. Convenient dates were secured to visit the community tribal halls for interviews.

A triangulation of data collection methods was applied to obtain broader and more detailed information on digital maternal and child health. For example, we adopted semi-structured and focus group interviews. During focus group discussions, participants could feed off each other as they responded to each other’s comments, sharing their ideas and perceptions, which was not evident during semi-structured interviews.

During individual semi-structured interviews, participants’ reflections were influenced by personal views and experiences. The first four authors conducted the interviews with an interview guide for two months (December and January 2021) using the locally spoken language of Sepedi. Interviews were conducted in sequence, starting with individual semi-structured interviews and focus groups with three different groups of participants (mothers, community leaders/fathers, and community healthcare workers).

#### 2.4.1. Semi-Structured Individual Interviews

Individual semi-structured interviews were conducted with six participants (two community leaders, two community health workers, and two mothers). The interviews were conducted using an interview guide to guide the proceedings of the interviews. Each interview lasted for 30 to 45 min, with COVID-19 safety regulations and privacy being maintained. Mothers who brought their babies were given adequate space to attend to the baby’s needs before proceeding with the interviews.

#### 2.4.2. Focus Group Activities

The focus group discussion was explained beforehand and only started when participants had given their voluntary consent to their participation. Two focus groups with six participants in each group (two community leaders, two community health workers, and two mothers) were conducted to obtain more in-depth information about using digital support apps for maternal and child health. More importantly, focus groups enabled the participants to provide a more detailed, in-depth description of the phenomenon and to identify shared knowledge and experience within a group [39]. The authors guided the focus group discussions while allowing the arguments from participants to flow. To avoid dominance bias during the focus group interviews, the authors started by outlining the purpose of the interviews and allocating equal time to each participant. Additionally, the authors also started by asking warm-up questions to break the ice and gain confidence and cooperation among the focus group members. The responses to the initial question guided the discussions. Discussions, which lasted for 1 h 30 min, were audio-recorded.

### 2.5. Data Analysis

Data collected through individual interviews and focus groups were analyzed using a thematic open coding method [30]. The team of three authors listened to audio recordings and transcribed them verbatim while translating them from Sepedi to English in preparation for data analysis. A qualitative and Sepedi expert verified the translated transcription. Three authors read through the transcripts and familiarized themselves with the data. After that, the authors generated initial codes from each data segment relevant to the experiences of maternal and child health digital support. The authors examined the codes in search of themes and grouped the findings into themes. Thereafter, a meeting was set up with other members of the research team who have experience in qualitative research to identify and refine the themes further. Data associated with each theme were reviewed and considered whether they supported the themes. Finally, the themes were defined and named as outlined in Table 2. [42].

### 2.6. Quality Assurance of Data

The following criteria were observed to ensure the accuracy of the results. Credibility was ensured by engaging with mothers and community health workers for prolonged interviews to obtain rich data. Transferability was ensured by providing a sufficiently thick description of the research process to give readers a proper understanding. This included the description of the number of participants who were purposively selected [31]. Furthermore, the study findings were compared and contrasted with the existing literature from a different context. To ensure dependability, participants’ reflections were transcribed to facilitate data analysis.

## 3. Results

The following themes in Table 2 emerged from the individual semi-structured interviews and focus groups conducted with mothers, community health workers, and community leaders in rural areas of Limpopo Province.

### 3.1. Theme 1: Reflections on Using Different Existing Digital Support Apps for Maternal and Child Health

Mothers and community health workers reflected on using different existing digital support apps for maternal and child health. For example, participants reported that the existing digital support apps enhance communication about maternal and child health issues. However, some participants expressed that existing digital support apps are inadequate for seeking maternal and child healthcare service assistance and that they currently preferred using traditional healers because of their cultural beliefs and practices. Therefore, the theme concludes by addressing the perceived usefulness of digital and non-digital ’resources’ for maternal and child health.

#### 3.1.1. Sub-Theme 1.1: Existing Digital Support Apps Enhanced for Communicating Maternal and Child Health Issues

Mothers described the existing digital support apps as those that enhanced communicating maternal and child health issues. For example, mothers shared the information conveyed to them about their pregnancy with other pregnant women, indicating how the app assisted them. Mothers reported that the whole pregnancy journey was full of pieces of advice from the MoMConnect app about pregnancy, duration of pregnancy, and expected dates for delivery, and mothers found this information to be useful. These findings are supported by the following illustrations:

Mother 2: *“About the application that assisted me during pregnancy, if I can mention a few, it will be on my WhatsApp…I was using Mom-Connect which assisted me during the whole process of my pregnancy”.*

CHW 3 added by saying *“it also helps when you are pregnant you know that at this time how far are you and when should you expect to give birth and as a mother you start to prepare for birth and buy clothing. It provides us with more information that we don’t know”.*

Mother 4 added *“since I have been pregnant I have been getting those messages from mom-connect advising me about the Pregnancy…After I gave birth I received a message…about going to the clinic for immunization and growth monitoring even now my child is one year, am still getting those messages”.*

#### 3.1.2. Sub-Theme 1.2: Perceived Strength and Weaknesses of Digital Support Apps on Mater-Nal and Child Health Services

Participants described the advantages and disadvantages, the useful and not useful of digital support apps on maternal and child health services. For example, mothers said that using MoMConnect felt like chatting with someone via SMS who provides updates about fetal development, antenatal visits, and follow-ups. Although access to MoMConnect app services required data, mothers expressed the benefits of using the app regarding fetal development and the updates that it gives on complications. Some mothers did report that data was a limiting factor for access to apps that needed data to run. The following quotes demonstrate these findings:

Mother 1 said, *“for mom-connect is like am chatting with a person, is like SMS, I just open it and communicate. For pregnancy plus, I must login to their system and access the services, and some of the services I cannot access without data”.*

Mother 4 reported that *“When I first came to the clinic, the sister took my cell phone number and registered me to mom-connect…mom-connect helps with ANC updates and visit dates”.*

Mother 5 reported that *“Mom-connect will notify me about my next ANC visit….and informs me about the baby’s development. They send a lot of information every week about child care e.g if a child is vomiting what you should do and if the child is not improving when you should go to the clinic.”*

#### 3.1.3. Sub-Theme 1.3: Preferences of Using Traditional Healers for Seeking Maternal and Child Health Care Service

Some participants expressed that existing digital support apps are not adequate for seeking maternal and child healthcare service assistance but preferred using traditional healers. Mother participants preferred seeking help from traditional healers about their child’s sickness and pregancy, which community health workers felt contributed to late antenatal bookings for pregnant women.

Mother 2 reported *“oh home, traditional I will start home because we believe in tradition. I have a traditional healer at home, so we can go to the clinic if the traditional healer fails. The traditional healer assesses if they could be able to assist with the maternal and child health problem, and if they ’can’t then, I will go to the clinic”.*

CHW 1 *“late pregnancy booking is a major challenge because they first consult with the traditional healers and come to the clinic when is already late”.*

CHW 2 added by saying *“Late pregnancy booking is a major challenge because they first consult with traditional healers and consult in the clinic when it’s already late”.*

#### 3.1.4. Sub-Theme 1.4: Role of Media Platforms in Advertising the Digital Support Apps for MCH

Participants reflected on media platforms such as TV, radio, and phones, which advertise the digital support apps for MCH. For example, mother participants preferred accessing maternal and child health information through the phone, which is always accessible as compared to the Road to Health booklet. They also expressed that they learned about MoMConnect from TV, which taught them about child care. Radio as a media platform also provided additional information about child care, such as managing diarrhea at home to prevent dehydration. The following quotes demonstrate the findings:

Mother 1 said, *“I heard one project on the radio about managing diarrhea, they were advising us on the home remedy so that the child does not become dehydrated and also on the signs to show that the child must be rushed to the hospital or clinic such as sunken eyes or fever”.*

Mother 2 said: *“I prefer to access information through the phone because unlike the road to health book which you put it away, but with the phone, we spend a lot of time using it even when I’m bored I read the information sent through mom-connect”.*

Community Leader 4 added, *“I have heard about mom-connect from the television what I know about it is that it assist mothers in taking care of the child through messages”.*

### 3.2. Theme 2: Challenges Experienced by Mothers about MCH Digital Support Apps

Participants reported challenges experienced by mothers with the digital support apps during the antenatal and postnatal periods. For example, mothers with smartphones said that app downloads are a challenge due to the lack of smartphones, data, and capacity on existing cell phones. The challenge downloading digital support apps was exacerbated by poor network connectivity and data affordability, which had a significant impact on the use of apps, and accessing MCH updates. The following sub-themes provide further discussion about this theme:

#### 3.2.1. Sub-Theme 2.1: Poor Network Connectivity and Data Affordability Affecting Usage of Existing Apps (for Mothers with Smartphones)

Two mother participants expressed severe concern with the network connectivity in the area and affordability, which influence the usage of digital support apps. Although some mothers receive messages from MoMConnect about their pregnancies, the lack of smartphones needed to access the app’s features due to network connectivity challenges remains a significant concern. Poor network connectivity was supported by community health workers participants who indicated that connecting from all network providers is a struggle. The following quotes below demonstrate these findings:

Mother 3 said, *“ Though I have used mom-connect during pregnancy, network in this area is a serious challenge”.*

Mother 6 said: *“ I don’t have a smartphone with the internet but receive mom connect messages”.*

CHW 1 said, *“it can work better if they can improve the network coverage and connection wherein even those who cannot afford to buy data can have a free connection and access the information.”*

Community leader 1: added *“I know nothing about the apps but what I know is that in this area network connection is a problem.”*

#### 3.2.2. Sub-Theme 2.2: App Downloads Are a Challenge Due to the Lack of Smartphones, Data, and Capacity on Existing Cell Phones

Participants reported challenges relating to the use of the MoMConnect app (National Department of Health, South Africa) and a lack of space required by the Pregnancy+ app (Philips Digital UK Limited, UK), which prevented access to updates about their pregnancies. For instance, four mothers experienced challenges with Aapp downloads due to the lack of smartphones, data, and capacity on existing cell phones. At the same time, one mother reported having failed to download Pregnancy+ due to a lack of space on the phone. The following quotes support these findings:

Mother 1: *“Pregnancy plus, it needs more space than mom-Connect because when it stops, its needs you to have another app and more space on the phone to accommodate another app”.*

Mother 2: *“I never used the app because I don’t have a smartphone, but I receive mom-connect messages since I have been pregnant I have been getting those messages advising me about the pregnancy and when to visit the clinic.”*

Mother 3: *“Reported that after delivery mom connects stopped…and I was asked to download pregnancy plus which I didn’t because of space”.*

Mother 4: *“I have been using Mom-connect, but sometimes I run short of data to use, but I still receive some messages.”*

#### 3.2.3. Sub-Theme 2.3: Poor Socio-Economic Status Leading to Lack of Adherence to Health In-Structions

Participants perceived poor socio-economic status as a leading factor in the lack of adherence to health instructions—for example, mothers reported that accessing the healthcare clinics for emergency child care was a struggle due to a lack of money. Mothers also said that a lack of finances resulted in them mixing formula with soft porridge, which was not the initial feeding plan. The following quotes support the findings:

Mother 1 reported, *“Sometimes I have an emergency for the child and when I ’don’t have money it becomes a challenge to reach the clinic, the ambulances take time to come as well.”*

Mother 2 added that *“I am using a formula feeding method, at times—I run short, which makes me use mixed with soft porridge.”*

CHW 5 said: *“the ambulance is not accessible as such I couldn’t take my child to the clinic, i don’t have medical aid for private doctors…its difficult to access health service when you don’t have money”.*

### 3.3. Theme 3: Promoting Digital Health Technologies of MCH Care

Participants suggested various measures to promote digital health technologies for MCH care. Digital health technologies for MCH care in rural areas could be promoted by ensuring a sustainable network of connectivity to access the internet. Furthermore, support groups on WhatsApp could be established to improve the sharing of information about MCH care. Lastly, participants suggested that information be shared at PHC waiting areas where you could have app recordings playing on TV screens. The current booklet-based information could be made available on voice/video recordings, allowing people to listen and watch. The following sections provide a detailed discussion about the theme:

#### 3.3.1. Sub-Theme 3.1: Designing an Integrated Single Digital App for Antenatal and Post-Natal Care with Communication Features

Mothers suggested that maternal care and complications could be improved by designing an integrated single digital app for antenatal and postnatal care with video call allowance. The single digital app with communication features such as video calling could assist mothers in communicating with their obstetrician to communicate pregnancy and postnatal complications. Mothers also wanted the app to function during pregnancy and postpartum periods to guide mothers about antenatal and postnatal care. The following illustrations of mothers support the findings:

Mother 1 said, *“We can overcome complications if they make pregnancy plus like mom-connect…to continue after term without having to download another app. After giving birth it must show a person that the baby is at one month and after one month, six weeks, it will indicate that it’s time for baby scale. And for pregnancy plus, its challenges can be overcome by making it continue without download as it was started”.*

Mother 2 added, *“Okay, we can improve, by, you know that most of the women we have complications, so at times I ’can’t go out to the clinics. But I thought we could add video calling, especially to talk about complications”.*

CHW 6 said *“It can be an App project for improving awareness about early bookings during pregnancy and child care after pregnancy”.*

#### 3.3.2. Sub-Theme 3.2: Primary Health Care Waiting Areas to Have App Recordings Played on TV Screens

Participants felt that having TV screens that continuously play the app’s recordings in the waiting areas of the primary healthcare settings could improve access to maternal and child health information—for example, having a video trend about pregnancy and child care can stimulate a father’s interest in maternal and child health. Another method of providing access to maternal and child healthcare information could include displaying MoMConnect content over TV screens to help those without smartphones. The following quotes support the findings:

CHW 1 said, *“The other thing, they can create awareness of the mom connect App by using the TV screen in the clinic to show them how it works isn’t it that they all wait there to be attended by the sisters”.*

Community leader 3: added by saying *“you see the video trend fast like the one for jokes and funny things if we can have the ones for pregnancy and child care I’m telling you even the fathers will watch when they circulate”.*

Mother 3 said, *“The clinic can give them recordings or let them listen to such recordings. These could be done by separating mothers from other community members in the waiting areas so that they can listen or watch the videos about pregnancy and child development”.*

#### 3.3.3. Sub-Theme 3.3: Improving Network Connectivity in the Rural Areas

Participants suggested that improving network connectivity in their area could be beneficial to both pregnant women and mothers. Mothers and community healthcare workers felt that having good network coverage and free network connectivity is a better way to provide access to maternal and child healthcare information. These suggestions are supported by the following quotes below:

Mother 4 said *“You can improve the network coverage and also access the apps in their homes or build a hall and advise them one-on-one they will understand better”.*

Community leader 5 added: *“improving coverage for and make the internet for free and also for us to advise them [pregnant women/mothers] to buy smartphones, we can also advise them to use their ”’children’s phones”.*

CHW 3: *“Mom connect can work better if they can improve the network connection wherein even those who cannot afford to buy data can have a free connection and access the information”.*

#### 3.3.4. Sub-Theme 3.4: Establish Support Groups on WhatsApp to Improve Uptake of the Ser-Vices and Share Information about the MCH Care

Participants expressed the desire to establish support groups on WhatsApp to improve the uptake of the services and share information about MCH care. The support group would allow pregnant women and mothers to discuss maternal and child health issues and create awareness about pregnancy bookings. Further benefits for the group will come from having the fathers share ideas about supporting mothers in their homes. These findings are supported by the quotes below:

Mother 1 said, *“I think having a support group on WhatsApp where we can talk about all these issues of maternal and child health.…”.*

CHW 4 added, *“It can be a support group amongst mothers to create awareness about near bookings during pregnancy because there are other conditions that can be detected and treated early during pregnancy, but if they are treated late, there is a possibility of losing both the mother and child or one of them”.*

Community leader 1 added: *“We can have more groups involving men wherein they can gather or share ideas as males using what’s-up or Facebook on how to assist the mothers in taking care of our children, women do that more often, but men we don’t”.*

#### 3.3.5. Sub-Theme 3.5: Information in the Road to Health Booklet Be Available on Voice/Video Recordings

The study findings acknowledge the benefits of the information in the ANC booklet; however, participants expressed that having this information recorded in videos/voice in the local language could be beneficial in spreading the MCH news quickly. Participants indicated that people are more attracted to videos than reading, so having recorded voice and video can spread the information fast. The following quotes support the findings:

Mother 1 said, *“You see, people are more attracted to videos than reading, if nurses can shoot a video about pregnancy or child health care, the information will spread fast through Facebook, whats-up and we can also share with those we know they are pregnant”.*

CHW 6 added by saying *“you see the video trend fast like the one for jokes and funny things if we can have the ones for pregnancy and child care I’m telling you even the fathers will watch when they circulate”*

Mother 3 said, *“Ooh, there is a lot of information in the booklet, so I think many people might fail to read such information, so making a recording about it and sending it to a phone, one can be able to listen to it during house chores”.*

Mother 6 said, *“Some of us cannot read well, but if there is music or play it is easier to listen on the phone…so I can listen to any MCH information on a phone any time I like as long as it is in my language”.*

## 4. Discussion

In line with the 2030 SDG 3 of improving good health and well-being, the SA National Digital Health Strategy 2019–2024 was developed to assist patients seeking access to healthcare services, healthcare workers to provide better services, and health systems managers to fulfill their roles, empowering all citizens to better navigate their personal health journeys using digital technologies [23]. This research is aimed at reflecting the perspectives of mothers, community leaders, and healthcare workers on the implantation of maternal and child health digital support in rural areas of South Africa. We have noted that mothers, community leaders, and healthcare workers utilize MomConnect, Pregnancy+, WhatsApp, and SMS for digital support services. MomConnect is a cell phone-based app that aims to register all pregnant women attending public health services while ensuring a healthy pregnancy and infanthood by sending targeted health promotion messages [7]. Pregnancy+ is a cell phone-based app that provides pregnant women with expert advice and healthcare tips for the baby’s development. WhatsApp and SMS provide cellphone-based communication services for pregnant women to share their experiences and challenges. Although the participants shared the benefits of using the digital support apps during pregnancy and infancy, there were challenges related to the utilization of the apps that could be improved for better utilization, for example, improving network connectivity and integrating MCH services such as Road to Health Chat with media platforms. The study further reflected on the role of traditional healers, which is perceived as a priority for seeking MCH services by most participants, thus emphasizing the current debate about integrating traditional practice into the healthcare system. The current study findings are aligned with the global commitment to curb the maternal and child mortality rate by 2030. We believe that empowering pregnant women and community health workers could make a significant contribution to lowering the high mortality rate in the rural areas of Limpopo Province.

### 4.1. Reflections on Using Different Existing Digital Support Apps for Maternal and Child Health

Mothers, community leaders, and health workers reflected on the use of existing maternal and child health apps such as Pregnancy+ and MoMConnect. Digital maternal and health apps (Pregnancy+ and MoMConnect) have an essential role in supporting MCH services such as receiving constant advice messages about pregnancy and fetal development, clinic and follow-up visits, and child care and development after birth. Study findings suggest the supportable use of MCH apps by mothers, community leaders, and healthcare workers in rural areas to ensure that all pregnant women and mothers enjoy the benefits brought by the two apps. The current study reveals the good experiences of participants with the MomConnect app and Pregnancy+, which gave first-time mothers learning opportunities about different pregnancy stages and their expectations throughout their pregnancy journey. Similar studies also reported that mothers who experienced pregnancy and child health challenges found the app messages comforting in their challenges with constant communication [7]. Different authors acknowledged that the MomConnect program creates insight and understanding regarding maternal and child care and, more importantly, boosts confidence [7,43,44]. Therefore, the use of Pregnancy+ and MoMConnect has the potential to enhance health instructions, especially with the chatroom opportunity between pregnant women or mothers and a health expert. The current study was conducted in a rural-based context where cultural norms and values play a significant role in people’s lives. Participants’ preferences on the use of maternal and child digital health indicated a major role that could be played by traditional healers in maternal and child health. Therefore, the study suggests that incorporating traditional healers when designing digital health apps can improve their usage and sustainability in rural areas.

### 4.2. Challenges Experienced by Mothers in Using MCH Apps during the Antenatal and Post-Natal Period

Despite the discussed essential roles of the maternal and child health apps, participants experienced challenges when using these digital support apps. For example, some mothers reported that it was challenging to access Pregnancy+ due to costly data. Such issues were also raised by Brusniak et al., who indicated that a lack of available data defeats the purpose of digital support apps in MCH care [45]. Participants also raised concerns that the Pregnancy+ app needs to be updated after 40 weeks. This is a challenge in terms of space capacity on devices, especially since some of the mothers had smartphones with limited storage capacity. The community experienced challenges with app downloads due to the lack of smartphones and data related to the findings of Hughson’s [46] study, which emphasized the importance of improving access to relevant and accurate information for the community and the mothers using pregnancy apps [46]. Our findings are in line with previous studies that have reported barriers to using different maternal and child health apps such as data cost, lack of equipment, and technology gaps [30,36].

The current study was conducted in rural areas of Limpopo Province, South Africa, where poverty is still a major challenge. The study results revealed some challenges related to poor socio-economic status, leading to a lack of adherence to health instructions. While some participants raised a concern that they cannot afford smartphones, it is also those same marginalized groups that cannot afford transport to reach services, which impacts access to MCH information and services. Therefore, our findings highlight the impact of socio-economic status on accessing MCH services either in person or virtually within rural areas. These findings are consistent with a study that deliberated on improving patient care for “unreachable populations” by overcoming the limitations imposed by cost and access, “digital health” or “mobile technologies” [45,46]. The current study’s findings are consistent with other scholarly work within the African continent, which raised concerns about the poor socio-economic status of pregnant women, which impacts the utilization of apps on the phone. For example, a study in Egypt on the health-seeking behavior amongst women from poor households recommended that quality provision of maternal health can be achieved by understanding the context, surroundings, and affordability among the poor and, more importantly, eliminating the inequality in maternal health coverage [47]. In South Africa, a study conducted by Scorgie, Blaauw, Dooms et al. on experiences of poverty and pregnancy found that transport costs to go to healthcare facilities were cited as a significant challenge experienced during pregnancy [48]. Our study findings are in contrast to other studies such as that by Hughson, Daly, and Wood-kron et al., which found that most urban women in the United States of America can afford smartphones that assist them with their pregnancy care and monitoring [45]. This is important because it highlights the need to think about context and the socio-economic environment when implementing the use of technologies in African contexts to support MCH and reveals that models that work in high-income settings likely need to adapt to the African context. More studies should be conducted to explore the status quo of pregnant women with smartphones accessing MCH apps in rural and urban areas of South Africa. Additionally, scholars have raised a need for the SA government to address the persistent problem of socially determined inequalities affecting maternal and child health outcomes [43,44]. For example, developing and implementing a national health insurance (NHI) scheme that aims to provide universal healthcare access for all while addressing inequalities in the provision and access to healthcare services will significantly contribute to MCH care services but will require the necessary resources to administer the scheme.

The current study findings observe the imbalance of socio-economic status for service access in rural areas, with participants reporting failure to buy data for their smartphones and poor network coverage in rural areas. Network providers could be involved in the solution across the country by improving the coverage and connectivity to satisfy end-users. Since connectivity is a problem, the SA National Digital Health Strategy 2019–2024 is in the process of addressing network connectivity constraints that hamper digital health efforts in South Africa [23]. This will be achieved by establishing the health network infrastructure with the relevant government departments to provide digital health broadband connectivity.

Furthermore, our findings create awareness among the program providers of digital support apps (MoMConnect and Pregnancy+) about the challenges experienced by pregnant women and mothers in deep rural areas. The awareness could assist the program designers in aligning the apps with pregnant women’s and mothers’ needs regarding MCH care. More importantly, the study points out the significance of potential collaboration between the Department of Health and the South African network providers to forge a good working relationship for integrating MCH services and digital support apps in rural areas. The study findings highlight the use of media platforms for MCH (e.g., phone, radio, and TV), providing an opportunity for integrating the media platforms with digital support apps for better MCH care services.

### 4.3. Promoting the Use of Digital Health Technologies

Participants raised various measures that could be used to promote digital health technologies for MCH care services in rural areas. The proposed measures aim to assist pregnant women and mothers during the antenatal and postnatal periods. They suggested that the questions and answers sections should be user-friendly and straightforward even after delivery, not requiring downloads. Moreover, it could also be user-friendly if the apps utilize low-cost data. Repeat sessions for MomConnect registrations were viewed as costly regarding user time and network charges. The issue of apps that require frequent downloads is consistent with findings reported in a study conducted in South Africa [31]. Moreover, they were concerned that the MomConnect message system is good at sending one-way messages from the center to the patient but not well designed to allow interaction [31]. Expensive data, messages, and poor network coverage were identified as barriers to the implementation and facilitation of digital health—the initiation of projects that could improve MCH services is welcomed by the participants, such as integrating media platforms such as radio and TV with digital support apps to promote maternal and child health. The findings are consistent with current studies in South Africa and other countries, where the idea is to ensure healthy babies and pregnancies through digital support that enforces early booking [48,49].

In the USA, a study by Alio and Lewis explored the community perspective on the role of fathers during pregnancy and indicated the significant role played by men in providing support to pregnant women and lactating mothers [50]. It is also part of our recommendations that we find it important for the fathers to be involved and trained in digital maternal and child health to assist in having a healthy pregnancy and healthy child growth.

## 5. Conclusions

The paper reflected on the dominant digital support for maternal and child health in rural areas. Despite various technological challenges raised by mothers, community leaders, and healthcare workers, digital support remains significant for improving the accessibility of information to mothers and community health workers regarding maternal and child health in rural areas. Therefore, improved data utilization and network connectivity could benefit mothers, community leaders, and healthcare workers regarding digital support using mobile apps (mobile health) such as MomConnect and Pregnancy+ in rural areas. The study concludes that digital support can effectively improve maternal and child health services if implemented considering the context and rural community needs. The recommendations, therefore, include implementing a zero-rated mobile app with a toll-free number for effective utilization of the apps by mothers and community health workers in rural areas. Furthermore, mothers without smartphones could benefit from accessing recorded clips about maternal and child health services. Lastly, the study confirmed the traditional practices of mothers in accessing maternal and child health. Therefore, the study recommends incorporating traditional healers when designing maternal and child health digital apps could improve their use in rural areas.

### 5.1. Strengths and Limitations of the Study

This qualitative study was conducted in three communities in Limpopo Province’s rural areas. The triangulation method was used to collect data, ensuring the data’s credibility and richness. The use of triangulation also created innovative ways to understand the experiences of mothers, community health workers, and leaders on digital health support. Consequently, the findings of this study cannot be generalized to the entire district or province. The researchers adhered to COVID-19 safety regulations during individual semi-structured interviews; however, communication barriers were encountered due to social distancing and wearing masks.

### 5.2. Contributing to the Body of Knowledge

The study findings provide a point of reference for the implementation of digital health in maternal and child health issues in rural areas.The study presents findings based in a rural context where diverse cultural activities play a significant role, thus suggesting that context should be considered when implementing maternal health services.Findings also provide a reference for monitoring the implementation of digital health (MomConnect and Pregnancy+) in maternal and child health services.

## Figures and Tables

**Table 1 ijerph-20-01842-t001:** Population size according to interview methods.

Participants	Age Ranges (Years)		Number of Children	Access to Smartphones	Focus Group 1	Focus Group 2	Individual Interviews	Total
Mothers’ children below 2 years	15–25 25–35 35–45	2 2 2	1 2 3	Yes (4) No (2)	2	2	2	6
Community health workers (CHW)	15–25 25–35 35–45 46–55	0 0 4 2	2 2 3	Yes (5) No (1)	2	2	2	6
Community leaders including their role as fathers	15–25 25–35 35–45 46–55	0 0 0 6	9	Yes (4) No (2)	2	2	2	6
								18

**Table 2 ijerph-20-01842-t002:** Themes and sub-themes reflecting the experiences of mothers and community health workers on digital maternal and child health support.

Themes	Sub-Themes	Participants Frequency
Theme 1: Reflections on using different existing digital support apps for maternal and child health.	1.1. Existing digital support apps enhanced the communication about maternal and child health issues.	3
	1.2. Strengths and weaknesses of digital support apps for maternal and child health services.	3
	1.3. Preferences for using traditional healers for seeking maternal and child healthcare services.	3
	1.4. Role of media platforms in advertising the digital support apps for MCH.	3
Theme 2: Challenges experienced by mothers during the antenatal and postnatal period	2.1. Poor network connectivity and data affordability affecting usage of existing apps.	4
	2.2. A lack of smartphones and capacity of using existing cell phones.	4
	2.3. Poor socio-economic status leads to a lack of adherence to health instructions.	3
Theme 3: Promoting the use of digital health technologies	3.1. Designing an integrated single digital app for antenatal and postnatal care with communication features.	3
	3.2. PHC waiting areas having apps recordings played on TV screens.	3
	3.3. Improving network aerials connectivity.	3
	3.4. Introduction of MHC peer support groups to improve uptake of the services and use of apps suggested.	4
	3.5. Information in the Road to Health booklet available on voice/video recordings.	6

## Data Availability

The research data supporting this publication are provided within this paper.

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
