# Peer review of "Reflections on Digital Maternal and Child Health Support for Mothers and Community Health Workers in Rural Areas of Limpopo Province, South Africa"

_ijerph, 2023, doi:10.3390/ijerph20031842_

Round 1

Reviewer 1 Report

1. The reviewed article touches on the important and forward-looking issue of applying new technologies to protect life and health. It is noteworthy that the research was conducted in a rural area. The research shows how big the difference is between urban and rural areas on every continent.

2. I would add to the finding that digital health support using mobile and digital technologies, such as Mom-Connect and WhatsApp, provide opportunities to improve maternal and child health care not only in low- and middle-income countries.

3.This is an important study when the authors cite the fact that a pregnant woman or newborn dies every 11 seconds somewhere in the world, accounting for 2.8 million deaths per year.

4. The introduction and context are correctly developed, although the focus could have been more on the continent of Africa, as there is data for that. But that’s how we got a broader context with a global dimension.

5. Qualitative methodology sufficiently fulfills scientific standards. In addition to quantitative research, there is a need for qualitative research that provides an opportunity to go deep into the research problem. While the understanding of the mothers’ situation has been deepened and adequately illuminated, it is necessary for the reader to expand on who the „community health workers” are. Do they have any medical or professional background? How are they selected? Do they have opportunities to take part in training in, for example, the correct use of digital media?

6. To solve the problem and show the important aspects on digital Maternal and Child health support for mothers and health workers in rural areas in Limpopo province, it is not enough to postulate to the government. It is necessary to scientifically elaborate on the status of fiber optics or cell phone masts to show the possibility of continuing development.

7. Finally, it is necessary to provide information on whether there are trained digital health support workers. It is not only in Africa that there are problems with this, as medical or IT studies are geared towards other tasks. Are these workers casual? Are they needed after short or longer training? From these answers will be seen the broader glories and shadows of the proposed system.

The article raises an important - life issue. I hope that the next ones will deepen the explored aspects of digital Maternal and Child Health support for mothers and health workers in rural areas.

Author Response

REVIEWER 1 COMMENTS

COMMENTS

CORRECTIONS

PAGE NUMBER

quantitative research, there is a need for qualitative research that provides an opportunity to go deep into the research problem. While the understanding of the mothers’ situation has been deepened and adequately illuminated, it is necessary for the reader to expand on who the „community health workers” are. Do they have any medical or professional background? How are they selected? Do they have opportunities to take part in training in, for example, the correct use of digital media?

This was a qualitative study. A brief description of community health care workers and their role was also outlined

Line 142-152

To solve the problem and show the important aspects on digital Maternal and Child health support for mothers and health workers in rural areas in Limpopo province, it is not enough to postulate to the government. It is necessary to scientifically elaborate on the status of fiber optics or cell phone masts to show the possibility of continuing development.

Corrected

Line 59-87

Finally, it is necessary to provide information on whether there are trained digital health support workers. It is not only in Africa that there are problems with this, as medical or IT studies are geared towards other tasks. Are these workers casual? Are they needed after short or longer training? From these answers will be seen the broader glories and shadows of the proposed system

Coorected

Line 101-113

Reviewer 2 Report

I think the following comments will be helpful for resubmission:

1. The abstract should be more specific on the method and results

2. The whole introduction section should be re-write. Especially the previous work-related this field  and the contribution of this work

3. The research design should be described by a figure or flowchart.

4. The study timeline and data collection method should be described in detail.

5. Authors should add the demographic table of the participants.

6. The data analysis technic should add more specifically for both quantitative and qualitative analysis

7. The discussion section should be more specific based on the research aims of this paper.

The major reason for rejection:

1. Poor research design (Especially considering the third-party technology Mom-Connect and WhatsApp as a mHealth tool), which has minimal contribution or significance to this work.

2. A small number of participants.

3. Insufficient data analysis of the quantitative result.

Author Response

REVIEWER 2 COMMENTS

Comments

Corrections

Page number

The abstract should be more specific on the method and results

Corrected

Line 33-41

The whole introduction section should be re-write. Especially the previous work-related this field  and the contribution of this work

Corrected

Line 82 to 94

114-133

157-172

The research design should be described by a figure or flowchart

The methodology has been thoroughly described. There is no need for flow diagram

The study timeline and data collection method should be described in detail

Corrected

Line 255-259

. Authors should add the demographic table of the participants.

Corrected

Table 1 221-213

2. A small number of participants.

This was a qualitative study.  the article has 18 participants which is deemed sufficient for qualitative study.

The data analysis technic should add more specifically for both quantitative and qualitative analysis

This was a qualitative study. we indicated step by step from interviews how the themes were generated, yet again, we report only qualitative study NOT quantitative

284-295

The discussion section should be more specific based on the research aims of this paper.

Corrected

we adopted a structural approach to the discussion to ensure that all discussion points related to the study findings and the purpose of the study.

Line 550-557, 653-658

Reviewer 3 Report

Dear authors. 

Congratulations about all your works of the study. I do think you pay a lot of efforts in achieve these results. However, there are many parts of the study should be clarified to make the study more reliable and scientific.  

Study aim:

The qualitative study was conducted to reflect the experiences of mothers, community leaders and community health workers on mobile health opportunities in the context of maternal and child health in rural areas of Limpopo Province, South Africa. 

Study Design: A qualitative design to study digital health support for mothers and community health workers about Maternal and Child Health in rural areas of Limpopo Province in South Africa. 

Participants: Participants including mothers above 18 years old with less than two children and at least experience in utilize any digital app during their pregnancy, community health workers and community leaders including the role of the fathers. Please explain why you choose these three different sources of participants in these study. There are only 18 participants from three different sources and do you think the case numbers sufficient to be reprehensive of all people in the rural area in south Africa?

Data Collection: Please clarify how you prevented some suggestive questions or statements by the observers and participants especially when your data were gathered during focus group activity. If one or two of the participants or authors who guided the discussion expressed more and firmly about their experience, other participants may also agree on their thoughts and let the results of the data more consistent. How you avoid such condition. 

Author Response

Comments

Corrections

Page number

Participants: Participants including mothers above 18 years old with less than two children and at least experience in utilize any digital app during their pregnancy, community health workers and community leaders including the role of the fathers.

Please explain why you choose these three different sources of participants in these study.

There are only 18 participants from three different sources and do you think the case numbers sufficient to be reprehensive of all people in the rural area in south Africa?

Corrected

Corrected

NB: South Africa has nine Provinces and the current study was conducted in Limpopo Province. The qualitative sample size is determined by data saturation when no new information emerges.

(NB Qualitative studies require a minimum sample size of at least 12 to reach data saturation

Vasileiou, K., Barnett, J., Thorpe, S. et al. Characterising and justifying sample size sufficiency in interview-based studies: systematic analysis of qualitative health research over a 15-year period. BMC Med Res Methodol 18, 148 (2018). https://doi.org/10.1186/s12874-018-0594-7

Line 217-236

Line 233-236

Data Collection: Please clarify how you prevented some suggestive questions or statements by the observers and participants especially when your data were gathered during focus group activity. If one or two of the participants or authors who guided the discussion expressed more and firmly about their experience, other participants may also agree on their thoughts and let the results of the data more consistent. How you avoid such condition. 

Line 278-282

Round 2

Reviewer 2 Report

Thank you for correcting the manuscripts. Though the current manuscript is improved, I think the following corrections will be needed to further improvement of this manuscript for this journal:

1. I commented in previous reviews that the introduction section should add the previous work-related field and the contribution of this field. I think the author can refer to the following work as a reference of the latest contribution of work in this field:

"Islam, ABM Rezbaul, et al. "A Mobile Health (mHealth) Technology for Maternal Depression and Stress Assessment and Intervention during Pregnancy: Findings from a Pilot Study." 2022 IEEE/ACM Conference on Connected Health: Applications, Systems and Engineering Technologies (CHASE). IEEE, 2022."

2. Though the authors have added a new table on participants' demography. But it will be better to have a participants' demography with their age, children's information, and so on. This will increase the research quality as we as the data analysis quality of this work.

Author Response

REVIEWER 2 TABLE OF CORRECTIONS  SECOND ROUND

COMMENT

CORRECTIONS

PAGE NUMBER

Thank you for correcting the manuscripts. Though the current manuscript is improved, I think the following corrections will be needed to further improvement of this manuscript for this journal:

1. I commented in previous reviews that the introduction section should add the previous work-related field and the contribution of this field. I think the author can refer to the following work as a reference of the latest contribution of work in this field:

"Islam, ABM Rezbaul, et al. "A Mobile Health (mHealth) Technology for Maternal Depression and Stress Assessment and Intervention during Pregnancy: Findings from a Pilot Study." 2022 IEEE/ACM Conference on Connected Health: Applications, Systems and Engineering Technologies (CHASE). IEEE, 2022."

2. Though the authors have added a new table on participants' demography. But it will be better to have a participants' demography with their age, children's information, and so on. This will increase the research quality as we as the data analysis quality of this work

Corrected

Corrected

Line 153-162

Line 216-217
